# Adaptation of *Plasmodium falciparum* to humans involved the loss of an ape-specific erythrocyte invasion ligand

William R. Proto [1], Sasha V. Siegel [1], Selasi Dankwa [2], Weimin Liu[3,4], Alison Kemp[1], Sarah Marsden[1], Zenon A. Zenonos[1], Steve Unwin[5,7], Paul M. Sharp [6], Gavin J. Wright[1], Beatrice H. Hahn [3,4], Manoj T. Duraisingh[2] & Julian C. Rayner [1,8]

*Plasmodium* species are frequently host-specific, but little is currently known about the molecular factors restricting host switching. This is particularly relevant for *P. falciparum*, the only known human-infective species of the *Laverania* sub-genus, all other members of which infect African apes. Here we show that all tested *P. falciparum* isolates contain an inactivating mutation in an erythrocyte invasion associated gene, *PfEBA165*, the homologues of which are intact in all ape-infective *Laverania* species. Recombinant EBA165 proteins only bind ape, not human, erythrocytes, and this specificity is due to differences in erythrocyte surface sialic acids. Correction of *PfEBA165* inactivating mutations by genome editing yields viable parasites, but is associated with down regulation of both *PfEBA165* and an adjacent invasion ligand, which suggests that *PfEBA165* expression is incompatible with parasite growth in human erythrocytes. Pseudogenization of *PfEBA165* may represent a key step in the emergence and evolution of *P. falciparum*.

[1] Malaria Programme, Wellcome Sanger Institute, Wellcome Genome Campus, Cambridge CB10 1SA, UK. [2] Department of Immunology and Infectious Diseases, Harvard T.H. Chan School of Public Health, Boston, MA, USA. [3] Departments of Medicine and Microbiology, University of Pennsylvania, Philadelphia, PA 19104, USA. [4] Department of Microbiology, University of Pennsylvania, Philadelphia, PA 19104, USA. [5] Chester Zoo, Chester CH2 1LH, UK. [6] Institute of Evolutionary Biology, University of Edinburgh, Edinburgh, UK. [7] Present address: School of Biosciences, University of Birmingham, Edgbaston B15 2TT, UK. [8] Present address: Cambridge Institute for Medical Research, University of Cambridge, Hills Road, Cambridge CB2 0XY, UK. Correspondence and requests for materials should be addressed to J.C.R. (email: julian.rayner@sanger.ac.uk)

*P*lasmodium falciparum emerged from a group of seven *Plasmodium* species that infect African apes, collectively referred to as the *Laverania* sub-genus[1–3]. The closest extant relative of *P. falciparum* is *P. praefalciparum*, a parasite that infects western lowland gorillas. All existing *P. falciparum* strains form a monophyletic clade within the *P. praefalciparum* radiation[4], suggesting that the jump to humans may have happened only once[1]. There is currently no evidence that ape *Laverania* species infect humans, even among populations living in close proximity to wild apes[5–7]. *Laverania* species appear to be largely ape species-specific, even when their hosts are sympatric[8], although in captivity these restrictions are not absolute[9]. This host specificity is in contrast to other primate-infective *Plasmodium* species such as *P. knowlesi*, which transitions readily between rhesus macaques and humans[10]. The cause(s) of *Laverania* host specificity are currently unknown but are important to understand, given that the breaching of such a host barrier led to the establishment of *P. falciparum* as a human parasite.

*Plasmodium* host specificity could be the result of incompatibilities at the vector-host, vector-parasite and/or host-parasite interfaces. Of these, host-parasite interactions, particularly during *Plasmodium* blood stages, have been most strongly implicated in *Laverania* species-specificity to date. *Laverania* core genomes are generally highly conserved, but multiple genes involved in erythrocyte invasion have been gained and lost between *Laverania* species[4,11], and in one case horizontal gene transfer has moved two essential erythrocyte invasion genes into an ancestor of *P. falciparum*[12]. One of these, PfRH5, has been under adaptive evolution during the diversification of the *Laverania*[13], and binds to its receptor Basigin in a host-specific manner[14].

PfRH5 is unlikely to be the only factor restricting *Laverania* blood stage interactions, as erythrocyte invasion is a complex process involving multiple receptor-ligand interactions[15]. The multi-gene erythrocyte binding-like (EBL) family plays a central role in this process, and three EBL ligands are known to recognise glycophorins, the major erythrocyte surface sialoglycoproteins – PfEBA175 recognises Glycophorin A[16], PfEBA140 recognises Glycophorin C[17,18], and PfEBL1 recognises Glycophorin B[19]. The role of other EBLs is less well understood and one, PfEBA165, does not appear to play any active role in invasion because the gene encoding it, *PfEBA165*, is a transcribed pseudogene in the reference *P. falciparum* 3D7 genome[20]. Interestingly, the inactivating frameshift found in *PfEBA165* in the 3D7 genome, as well as the genomes of several other lab isolates, is absent in the chimpanzee parasite *P. reichenowi*[21] as well as other *Laverania* species[4,11,12] raising the possibility that inactivation of *PfEBA165* played a role in the successful colonization of humans by *P. falciparum*.

Here we use a combination of sequencing, protein expression, and red blood cell engineering to show that selective sialic-acid binding of EBA165 is a key determinant of host-specific erythrocyte binding. In addition, we find that CRISPR-Cas9 mediated correction of the inactivating *PfEBA165* mutations in *P. falciparum* silences expression of both *PfEBA165* and other genomic regions, suggesting that expression of functional PfEBA165 is not compatible with efficient growth of *P. falciparum* in human erythroyctes. Together, these data suggest that inactivation of *PfEBA165* may have been important in the emergence of *P. falciparum* as a human pathogen, and has broader implications for the role of invasion ligand changes in the ability of *Laverania* parasites to transmit between, and adapt to, new hosts.

## Results

**EBA165 frameshifts are restricted to human-infective parasites.** All *P. falciparum* EBL proteins have a conserved structure, with a signal sequence and large ectodomain encoding two erythrocyte binding domains, followed by a transmembrane domain and short cytoplasmic domain (Fig. 1a). Two frameshifts within *PfEBA165* were identified in the reference *P. falciparum* 3D7 strain[20], with the 5′ most frameshift occurring immediately downstream of the signal sequence, and the 3′ frameshift

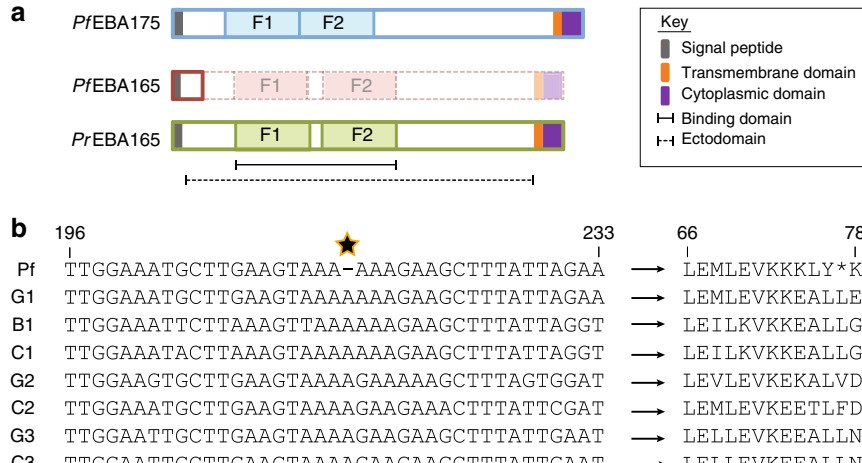

**Fig. 1** EBA165 frameshifts are specific to human-infective *Laverania* parasites. **a** Cartoon of EBL protein structure and domains. The erythrocyte binding domain of each protein is split into two Duffy Binding Like (DBL) erythrocyte binding domains (F1, F2). Bold outlines indicate translated amino acid sequence. In *P. falciparum*, the presence of a frameshift mutation eliminates translation of the majority of the protein, including both DBL domains, unlike the homologous protein in *Plasmodium reichenowi* (PrEBA165) or the major *P. falciparum* glycophorin binding protein, PfEBA175. The dashed/faded region of PfEBA165 depicts the protein structure that would be produced if frameshifts are corrected. **b** Alignment of *Laverania* EBA165 orthologues spanning the single base pair deletion (starred) universally conserved in 2517 globally distributed clinical *P. falciparum* isolates. Sequences displayed are from one isolate for each *Laverania* species (G1, *P. praefalciparum*; C1, *P. reichenowi*; B1, *P. lomamiensis*; G2, *P. adleri*; C2, *P gaboni*; G3, *P. blacklocki*; C3, *P. billcollinsi*). All seven non-human-infective *Laverania* species lacked the single nucleotide deletion (left hand side), eliminating the premature stop codon present in *PfEBA165* and resulting in intact amino acid sequences (right hand side). Residue numbers are indicated above the alignments and are based on the PfEBA165 sequence. Partial alignments of ape *Laverania* PfEBA165, along with a frameshift corrected *P. falciparum* sequence, are shown in Supplementary Fig. 1

occurring within the F1 domain (Fig. 1a). Analysis of *PfEBA165* in other lab strains showed the 5′ frameshift, which truncates the transcript before the erythrocyte binding domains, was conserved, while the 3′ frameshift appeared restricted to 3D7[20,21]. To establish the global frequency of these frameshifts we analysed *PfEBA165* sequences from 2517 *P. falciparum* clinical isolates using data from the Pf3k project (https://www.malariagen.net/projects/pf3k). The 3′ frameshift present in 3D7 was found to occur in only one other isolate, whereas all 2517 isolates, collected from across Africa, Asia, Oceania and South America, contained the identical 5′ frameshift. All *P. falciparum* strains therefore appear to lack an intact *PfEBA165* gene.

By contrast the genome of the closest living relative of *P. falciparum*, the gorilla-infective parasite *P. praefalciparum*, encodes a complete *PfEBA165* open reading frame, as do all of the five other ape *Laverania* parasites that have been whole genome sequenced to date[4,11,12]. To provide further evidence that inactivation of *EBA165* is *P. falciparum*-specific, we examined the frameshift containing region in additional ape *Laverania* samples. Using single template PCR to amplify a ~800 bp fragment from ape faecal samples collected in the wild, we identified 78 distinct *EBA165* haplotypes representing all seven ape *Laverania* species, including the recently discovered bonobo parasite *P. lomamiensis*[3]. None of these contained the *P. falciparum*-specific frameshift, or any other inactivating mutation within the amplified region (Fig. 1b and Supplementary Fig. 1).

**Laverania EBA165 selectively binds chimpanzee erythrocytes**. Given an intact *EBA165* open reading frame is only found in ape-infective *Laverania* species, we hypothesized that the encoded protein performs an ape-specific function during erythrocyte invasion. To test this we expressed the full-length ectodomain of *P. reichenowi* EBA165 (PrEBA165) as a recombinant protein, as well as a corrected version of PfEBA165, representing the *P. falciparum* sequence as it would be translated if no frameshift were present. Monomeric PrEBA165 and PfEBA165 expressed as soluble recombinant proteins of the expected size in HEK293E cells, as detected by immunoblot (Fig. 2a). PfEBA175, which we have previously shown to be functionally active when expressed in this system[22], was used as a positive control. Biotinylated monomeric recombinant proteins were immobilised on streptavidin-coated Nile Red fluorescent beads and incubated with either human or chimpanzee erythrocytes. After washing, the binding of protein loaded beads to erythrocytes was detected by flow cytometry (Fig. 2b). As expected, PfEBA175 bound both human and chimpanzee erythrocytes (Fig. 2c), and binding was eliminated by treatment with neuraminidase (dashed line, Fig. 2c), which enzymatically removes sialic acids from the erythrocyte surface, and is known to eliminate interaction between PfEBA175 and GYPA[16]. Both PrEBA165 and PfEBA165 bound to chimpanzee erythrocytes and their binding was also eliminated by neuramindase treatment, suggesting that like PfEBA175, binding of EBA165 proteins is sialic-acid dependent. By contrast, neither PrEBA165 nor the corrected PfEBA165 proteins were able to bind to human erythrocytes, suggesting that they can only recognise chimpanzee erythrocytes.

**Erythrocyte surface sialic acids dictate host specificity**. Given that binding of PrEBA165 and PfEBA165 was sialic-acid dependent, we investigated their binding preferences for sialic-acid variants, which are known to differ between humans and other apes. The *CMAH* gene that encodes the enzyme that converts *N*-acetylneuraminic acid (Neu5Ac) to *N*-glycolylneuraminic acid (Neu5Gc) was disrupted during the evolution of modern humans[23]. Human sialoglycoproteins therefore contain only

Neu5Ac, while ape sialoglycoproteins contain both Neu5Gc and Neu5Ac, with Neu5Gc representing approximately 75% of the sialic-acid presented on the surface of chimpanzee erythrocytes[24]. PrEBA165, PfEBA165 and PfEBA175 were expressed as pentamers, to increase binding avidity, and tagged with β-lactamase to enable detection. Consistent with previous findings[22,25] PfEBA175 was able to bind to a range of sialic-acid containing glycans, including both Neu5Gc and Neu5Ac sialic-acid variants (Fig. 3a; full glycan panel shown in Supplementary Fig. 2). By contrast, both PrEBA165 and the corrected PfEBA165 bound only to glycans containing Neu5Gc sialic acids (Fig. 3a; full glycan panel shown in Supplementary Fig. 2), mirroring their specificity for ape erythrocytes.

To test whether sialic-acid variants are the sole determinant of host-specific erythrocyte binding, we expressed *CMAH* in human haematopoietic stem cells (HSCs), then differentiated them to produce cultured human red blood cells (cRBCs) that have Neu5Gc containing sialoglycoproteins on their surface[26]. cRBCs differentiated from human HSCs transduced with a lentiviral vector containing chimpanzee *CMAH*, or an empty vector control (pLVX), had no detectable differences in their surface levels of a range of erythrocyte receptors used by *P. falciparum* for erythrocyte invasion (Glycophorin A, Glycophorin C and Basigin; Supplementary Fig. 3). Expression of *CMAH* also had no effect on PfEBA175 binding (Fig. 3b), which as shown above is able to bind both human and ape erythrocytes, and to Neu5Ac and Neu5Gc containing sialic acids. By contrast, expression of *CMAH* had a radical effect on PrEBA165 and PfEBA165 binding. Both were unable to bind to control cRBCs (pLVX, Fig. 3b), just as they had been unable to bind to mature human erythrocytes taken from circulation. However, both PrEBA165 and PfEBA165 could bind cRBCs differentiated from *CMAH* expressing HSCs, which differed only in the sialic-acid variant that was expressed on the red blood cell surface. PrEBA165 and the corrected PfEBA165 are therefore unable to bind human erythrocytes solely because of the absence of ape-specific Neu5Gc sialic acids.

**Correcting PfEBA165 frameshifts eliminates gene expression**. The ape-specific binding of corrected PfEBA165 suggests that its activity would not have been needed after the transition of ancestral *P. falciparum* parasites to humans. This would in turn have removed any selection pressure against acquiring inactivating frameshift mutations in *PfEBA165*, such as the one seen in all currently circulating *P. falciparum* parasites. To reverse this process we used CRISPR-Cas9 genome editing to correct the frameshift mutations. To allow modification of frameshifts singly or together, we used two separate sgRNAs and two linear PCR products that were engineered to correct either the 5′ or 3′ frameshift (Fig. 4a); transfection together yielded strains with both frameshifts corrected (C10 and D3), while transfection of only one yielded a strain with the 3′ frameshift corrected, but the 5′ frameshift intact (YB4). The presence of the engineered mutations was confirmed by PCR and Sanger sequencing (Fig. 4b). Whole-genome sequencing identified only three other non-synonymous SNPs in the edited lines relative to the parental line, which may have arisen during mitotic replication. None of the single and double-edited strains had any apparent growth defect under standard in vitro conditions, or in competitive growth assays (Supplementary Fig. 4A). Invasion phenotypes were explored using enzymes to remove subsets of erythrocyte receptors, which can reveal subtle changes in invasion pathway usage[27–29]. A trend of reduced invasion was observed for the double-edited lines when human (Supplementary Fig 4B) and chimpanzee (Supplementary Fig 4C) erythrocytes were pre-treated with neuramidase. Neuraminidase selectively removes

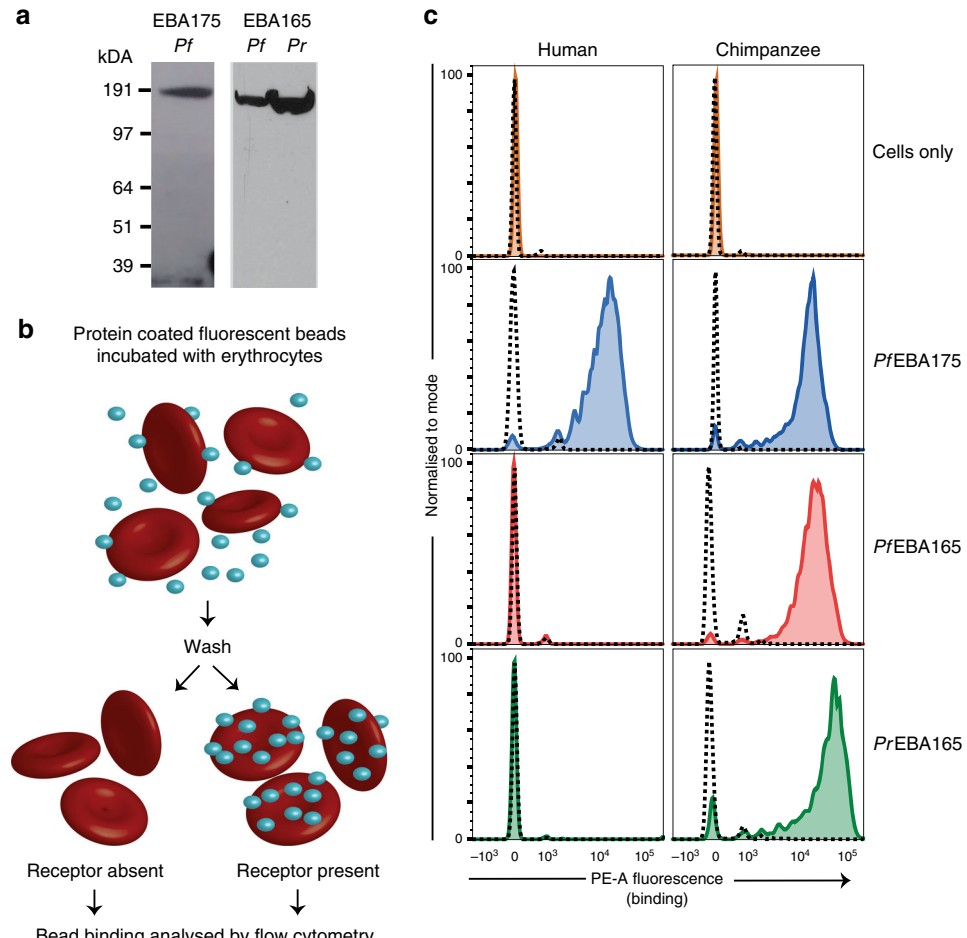

**Fig. 2** PfEBA165 and PrEBA165 bind chimpanzee, but not human, erythrocytes. **a** Recombinant PfEBA165, PrEBA165 and PfEBA175 ectodomains were expressed as secreted mono-biotinylated proteins in HEK239E cells. Immunoblots of HEK293E cell supernatant were probed with HRP-linked streptavidin to confirm expression of a single band of the expected sizes (PfEBA175: 188 kDa; PfEBA165: 177 kDa; PrEBA165: 177 kDa). The blot images displayed have been cropped and Source Data are provided as a Source Data File. **b** Cartoon outline of erythrocyte binding assays. Enzymatically mono-biotinylated proteins were captured on streptavidin-coated fluorescent beads, and incubated with either human or chimpanzee erythrocytes. After repeated washing, binding of fluorescent beads to the erythrocytes was detected using flow cytometry. **c** Erythrocyte binding assay using human and chimpanzee red blood cells incubated with PfEBA165, PrEBA165 and PfEBA175. Solid coloured lines indicate traces after binding to untreated erythrocytes, showing that PfEBA175 binds to both human and chimpanzee erythrocytes, whereas PfEBA165 and PrEBA165 bind only to chimpanzee erythrocytes. In all cases, binding is abrogated by neuraminidase treatment (dashed lines), which removes extracellular sialic acids. Assays were conducted in triplicate with the same batches of proteins and erythrocytes. Individual assays included three technical replicates for each condition tested; a single representative technical replicate is shown

extracellular sialic acids, suggesting that these lines have an increased reliance on sialic-acid dependent-invasion pathways. The other enzyme treatments tested did not affect invasion rates (Supplementary Fig. 4D).

Given the lack of a strong phenotype in any of the invasion assays, we used qRT-PCR to investigate whether frameshift correction had affected *PfEBA165* transcript levels (Fig. 4c). Clone YB4, where only the 3′ frameshift had been repaired, expressed similar levels of *PfEBA165* to the wild-type 3D7 control. By contrast, the two double-edited clones had negligible levels of *PfEBA165* transcript, suggesting that *PfEBA165* was specifically down-regulated when both frameshift mutations were reversed. To establish whether this downregulation was specific to *PfEBA165*, we performed strand-specific RNAseq. Differential expression analysis identified a total of 444 differentially expressed genes between 3D7 and C10, and 2202 between 3D7 and D3, however the number of genes with large log2 fold-changes was considerably smaller (36 and 359, respectively). Once known clonally-variant genes were removed, only a handful of significantly differentially expressed

genes remained (Fig. 5a). Interestingly, these genes were located in two large regions (~43 kb and 23 kb, respectively) on chromosomes 4 and 11 (Supplementary Table 1). The large downregulated region on chromosome 4 spans several genes, including *PfEBA165* and another invasion gene *PfRH4*, along with six additional genes downstream of *PfEBA165* (Fig. 5b). *PfEBA165* had log₂FCs of ~−2.2 in both clones compared to 3D7, consistent with the qRT-PCR data. The three most tightly down-regulated genes in both clones, sporozoite and liver stage asparagine-rich protein (SLARP, Pf3D7_1147000), a putative dynein light chain, (Pf3D7_1147100), and a putative tubulin tyrosine ligase (Pf3D7_1147200), are all located within a 23 kb region on chromosome 11 (Fig. 5c). *PfEBA165* frameshift editing therefore appeared to be accompanied by relatively large epigenetic silencing events, including down-regulation of *PfEBA165* transcription.

## Discussion

In this work we show that *PfEBA165* is a pseudogene in all circulating *P. falciparum* parasites, but the homologous gene is

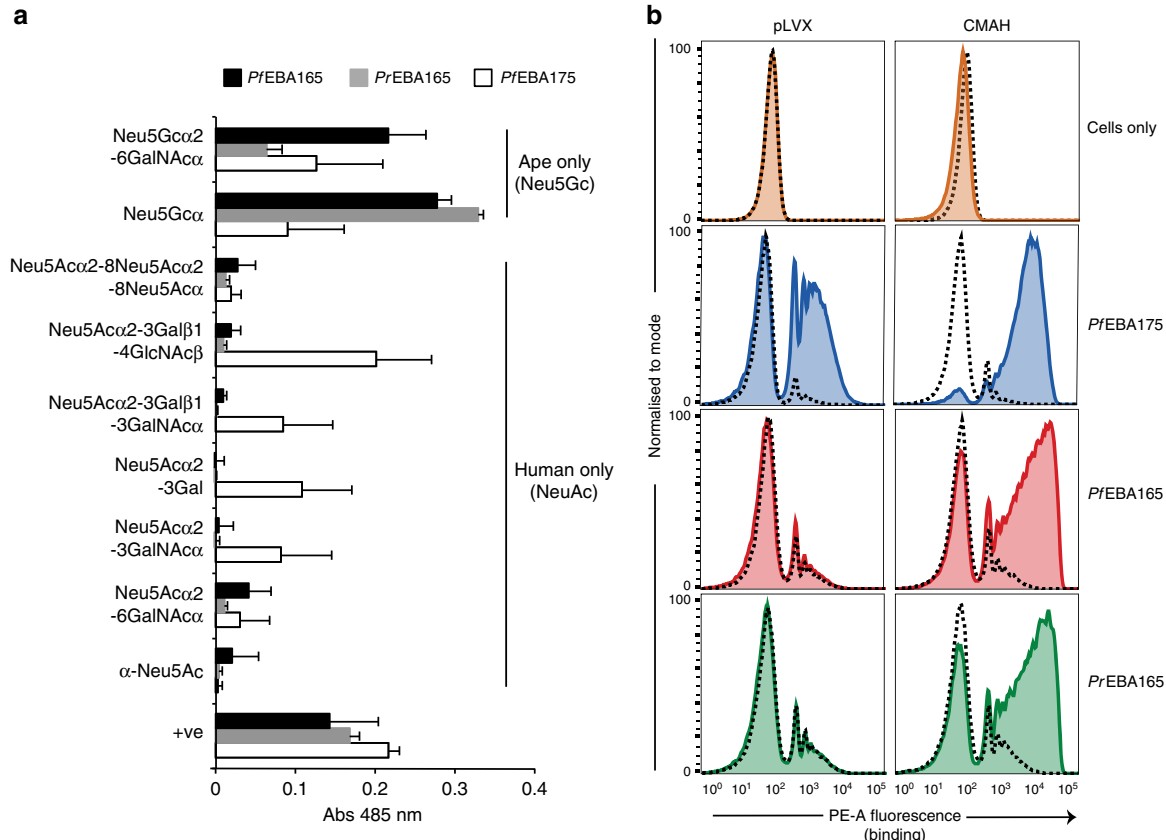

**Fig. 3** PfEBA165 and PrEBA165 binding is specific for ape sialic acids. **a** PfEBA165 and PrEBA165 bind ape, but not human, sialic acids. Purified mono-biotinylated glycans (GlycoTech) were immobilised on streptavidin-coated 96-well plates, and probed with normalised recombinant pentameric β-lactamase-tagged PfEBA165, PrEBA165 and PfEBA175 (red, green and blue bars, respectively). Binding was detected by incubation with nitrocefin, a substrate for β-lactamase. The full range of glycans probed are shown in Supplementary Fig. 2; only glycans containing sialic acids are shown here. PfEBA175 was able to bind to glycans containing both Neu5Ac and Neu5Gc, whereas PfEBA165 and PrEBA165 bound only to Neu5Gc containing glycans, which are specific for ape cells. Data points represent mean values of three technical replicates, conducted on the same batches of recombinant proteins and glycans. Error bars represent standard deviation. +ve = positive control, Ox68-bio. Source Data are provided as a Source Data File. **b** Erythrocyte binding assay using cRBCs incubated with PfEBA165, PrEBA165 and PfEBA175. Solid coloured lines show traces after binding to untreated cRBCs, indicating PfEBA175 binds to both cRBCs produced from both control and CMAH transfected hSCs. PfEBA165 and PrEBA165 bind only to cRBCs produced from CMAH transfected hSCs. In all cases, binding is removed by neuraminidase treatment (dashed lines), which removes extracellular sialic acids. Binding assays were repeated with two biological replicates of cRBCs and recombinant proteins from the same batch. Individual assays included three technical replicates for each condition tested; a single representative technical replicate is shown

intact in all ape *Laverania* species. Recombinant *Laverania* EBA165 ectodomains bind specifically to chimpanzee but not human erythrocytes, and this binding specificity is due to differences between the sialic-acid repertoire of human and ape erythrocytes. The human genome contains an insertion in the *CMAH* gene, resulting in the production of only Neu5Ac sialic acids, whereas apes contain predominantly Neu5Gc[23]. Sialic acids are a common component of cell surface glycans, particularly on erythrocytes where they decorate sialoglycoproteins such as Glycophorin A, which is present in $\sim 1 \times 10^6$ copies on every erythrocyte[30]. This change in sialic-acid composition between humans and apes has been implicated in a wide range of pathogen binding specificities, including recognition of macaque erythrocytes by *P. knowlesi*[26]. In the case of both PfEBA165 and PrEBA165 the binding specificity for Neu5Gc is absolute, as we show that adding a functional CMAH is all that is required to allow binding of these proteins to human erythrocytes.

Given that the ape *Laverania* EBA165 proteins tested bind exclusively to an ape-specific sialic-acid (Neu5Gc), there appears to be no requirement for *P. falciparum* to produce a functional PfEBA165 protein. However, it is striking that >2500 clinical isolates sequenced to date all contain exactly the same

inactivating mutation within *PfEBA165*, suggesting that this mutation was either present in the gorilla precursor to *P. falciparum*, or was acquired very early during the emergence of *P. falciparum* and selected to fixation. Using CRISPR-Cas9 genome editing to engineer a correction of the frameshift mutations in *P. falciparum* parasites led to an extreme downregulation of *PfEBA165* gene transcription, as well as a broader effect on genes in the vicinity such as *PfRH4*. Silencing of *EBA165* is consistent with previous studies indicating epigenetic control of invasion ligands, as *P. falciparum* differentially employs a suite of interchangeable ligands[31,32]. *PfEBA165* and *PfRH4* are known to share a bidirectional promoter, with co-regulation of *PfEBA165* and *PfRH4* observed in both lab[33] and clinical isolates[34,35]. PfRH4 binds Complement Receptor 1 and represents an important sialic-acid independent invasion pathway for *P. falciparum*[36,37]. The observed downregulation of *PfRH4* in the double-edited strains is likely to be responsible for the observed change in erythrocyte invasion pathway towards sialic acids dependent-invasion, perhaps through increased reliance on EBA175. The significant role of RH4 in invasion suggests that downregulation of this locus may have a cost in natural infection, and that even in vitro *P. falciparum* goes to considerable lengths to avoid

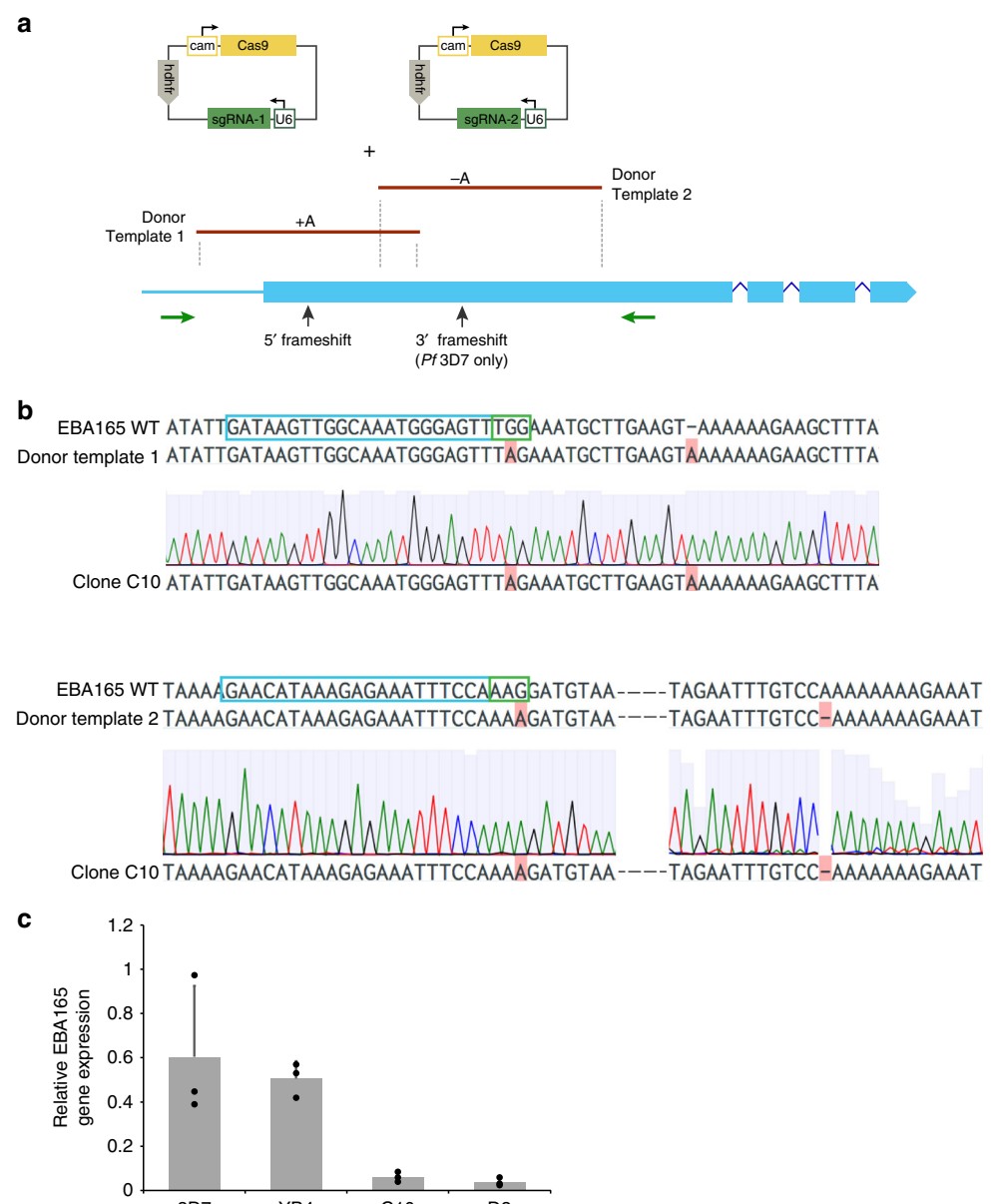

**Fig. 4** *PfEBA165* frameshift correction results in down-regulated transcription. **a** Schematic showing genome editing strategy. *P. falciparum* 3D7 parasites were transfected with plasmids containing Cas9 expression cassettes and sgRNAs targeting near the two *PfEBA165* frameshifts present in the endogenous gene, along with linear PCR donor templates containing corrected versions of these frameshifts. Cas9 mediated homologous recombination would result in correction of either one or both frameshifts, depending on whether one or both sgRNA containing plasmids were transfected. The plasmids contained the human dihydrofolate reductase (*hDHFR*) gene for selection, Cas9 and sgRNA expression were driven by endogenous calmodulin (cam) and U6 promoters, respectively, **b** Sanger sequencing of cloned double transfectant parasite line (C10) confirms correction of both frameshifts, with correction for 5′ frameshift shown in top row and 3′ frameshift shown below. Blue boxes outline sgRNA target sequences and green boxes outline PAM sites within the target sequence. The donor template incorporated changes that eliminate the PAM site, preventing re-cleavage of corrected genes, but do not affect the translated amino acid sequence. **c** Quantitative RT-PCR of *PfEBA165* expression in 3D7 (parental line). YB4 (cloned transfectant with the 3′ frameshift corrected) and C10 and D3 (cloned transfectant lines with both the 5′ and 3′ frameshifts corrected). To control for differences in relative abundance of mature schizonts in the parasite culture before RNA extraction, *PfEBA165* transcript levels were normalized to *Pfcyp87* (stable expression throughout cell cycle) and expressed relative to *PfAMA1* (late-stage specific expression). Data points represent mean values of three biological replicates. Error bars represent standard deviation and statistical significance was determined using a one-way ANOVA with Dunnet's multiple comparison test for each erythrocyte category. **$p < 0.01$. Source Data are provided as a Source Data File

expressing intact PfEBA165. Together, these observations indicate that expression of functional PfEBA165 protein is incompatible with successful growth of *P. falciparum* in human erythrocytes.

While this is an intriguing hypothesis, it was not possible to confirm. Multiple attempts to conditionally express an intact *PfEBA165* gene episomally were not successful, perhaps due to leakiness in conditional systems. No other *Laverania* parasites have been adapted to in vitro culture, so the complementary experiment of introducing an *EBA165* inactivating mutation into an ape *Laverania* genome is not possible. Finally, using cRBCs expressing *CMAH* to select for EBA165-expressing *P. falciparum* transfectants is also not possible, in this case due to technical

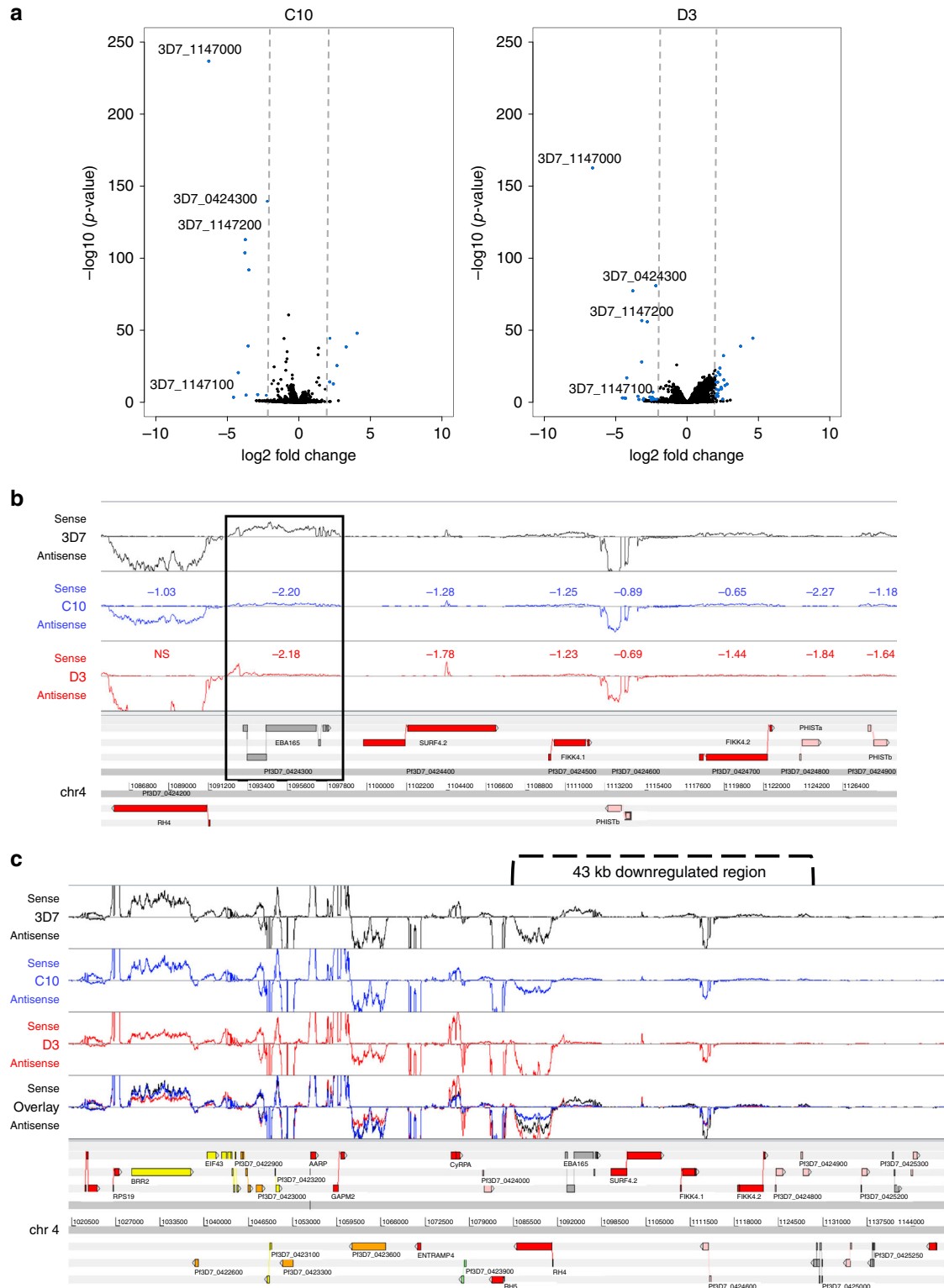

**Fig. 5** *PfEBA165* frameshift correction reveals clusters of down-regulated genes. **a** Volcano plots showing significantly upregulated and down-regulated genes in the *EBA165* double-edited parasite clones C10 and D3 relative to 3D7, plotted as -$\log_{10}$(pvalue) against $\log_2$ fold change ($\log_2$FC). Blue dots represent highly up-regulated and downregulated genes ($-2 > \log_2$FC > 2, adjusted *p*-value ≤ 0.05, DESeq2 Wald-test). Hashed lines indicate cutoffs of $-2 > \log_2$FC > 2. Dots with no gene IDs listed beside them are genes belonging to gene families that are known to be clonally variant in expression. **b** The genomic region on chromosome 4 that is downregulated in the *PfEBA165* restored lines. $\log_2$ fold change in expression relative to the wild-type 3D7 line is shown numerically above each gene; *PfEBA165* is highlighted in the boxed region. **c** The genomic region on chromosome 11 that is down-regulated in the *PfEBA165* corrected lines. $\log_2$ fold change in expression relative to the wild-type 3D7 line is shown numerically above each gene

limitations. While CMAH cRBCs can be produced at a small scale for single cycle invasion or growth assays such as those performed here, it is not currently possible to produce these cRBCs at a sufficient scale to support long-term cell culture experiments such as *P. falciparum* transfection or selection.

Given the clear ape-specificity of *Laverania* EBA165 binding, and our inability to express functional *PfEBA165* in *P. falciparum* parasites, it is interesting to speculate whether inactivation of *PfEBA165* might have been required for the ancestor of *P. falciparum* to successfully colonise humans. How might gene inactivation promote species transition? PfEBA165 belongs to a group of invasion ligands, the EBLs, which along with another multigene family of invasion ligands, the RBLs, are thought to play partially overlapping functions during invasion[38]. Deletion of one gene within these families often shifts the invasion process so that it relies on pathways encoded by other ligands[29] and such pathways are thought to be organised hierarchically[39]. If PfEBA165 was the dominant ligand in the invasion hierarchy of the gorilla-infective common ancestor of *P. falciparum* and *P. praefalciparum*, then the loss of that dominant ligand could have revealed the activity of a secondary invasion pathway that was not ape-specific (Fig. 6). Given what we know about currently circulating *P. falciparum* strains, it could have been PfEBA175 that became more important following the loss of PfEBA165, thus predisposing the ape-infective ancestor to be able to infect human erythrocytes. Under this theory, our attempts to recreate a functional *PfEBA165* reinstated that ape-specific dominant pathway, outcompeting the ability of PfEBA175 to function, and thus could only be tolerated if expression of the corrected *PfEBA165* gene was significantly down-regulated. Loss of *PfEBA165* may not have been absolutely required for species transition, but may instead have conferred a significant growth advantage in human erythrocytes once the initial jump had occurred, meaning that strains with the single inactivating mutation were selected for and rapidly outcompeted strains with a functional *EBA165* locus. Loss of invasion ligands have been linked with changes in host preference during the in vitro adaptation of *P. knowlesi* to growth in human erythrocytes. In this case, deletion of an RBL invasion ligand (PkNBPXa) and an EBL ligand (PkDBPgamma) were associated with increased invasion efficiency for macaque erythrocytes, and decreased invasion efficiency for human erythrocytes[26,40].

Of course, interactions between *Plasmodium* parasites and humans are complex and occur at multiple stages, so an intact *EBA165* gene is unlikely to be the only factor preventing ape *Laverania* species from infecting humans. In particular, there is compelling evidence suggesting that the changes to the PfRh5 invasion complex may also have played a crucial role. Completion of the *P. gaboni* genome revealed a horizontal gene transfer event which transferred Rh5 and CyRPA from the distantly related gorilla parasite *P. adleri* into the gorilla precursor of *P. falciparum*[12], which may have predisposed *P. praefalciparum* to colonize humans. Moreover, *Laverania* RH5 proteins have been under adaptive evolution, with one change occuring at the origin of *P. falciparum*[13], which may explain why PfRh5 appears to bind to its receptor Basigin in a host-specific manner[14]. Given that it is now clear that PfRh5 functions as part of a multi-protein complex[41–43], the species specificity of interactions between complex members warrants further investigation. However, the data presented here suggests that the introduction of a frameshift mutation in *EBA165* may have also played a key role in the emergence of *P. falciparum* as a human-infective species, being either required for the host transition from gorillas to take place, or necessary for increasing blood stage multiplication rates once the transition had occurred. The last decade has seen a rapid expansion in our understanding of the prehistory of

*P. falciparum*. This and other molecular studies are starting to reveal the exact steps that were required for *P. falciparum* to emerge as one of the most significant pathogens in human history.

## Methods

**Recombinant protein production.** *PfEBA175*, *PfEBA165* and *PrEBA165* sequences were obtained from published reference genomes[11]. PfEBA175 and PfEBA165 recombinant proteins were produced by transient transfection of HEK293E cells[44]. Briefly, protein ectodomains were identified using signal peptide and transmembrane helices prediction software[45,46]. Potential N-linked glycosylation sites (NXS/T sequence motifs, X is any amino acid except proline) were removed by replacing serine or threonine residues with alanine, because *Plasmodium* species don't naturally glycosylate secreted proteins, but cryptic sites can be inappropriately glycosylated by other eukaryotic expression systems. Coding sequences, flanked by unique *Not*I and *Asc*I restriction sites, were codon optimised for expression in human cells and were produced by gene synthesis (GeneArt). To produce biotinylated monomeric protein, modified ectodomains were cloned into a derivative of the pTT3 expression vector, between an N-terminal signal peptide (mouse variable κ light chain 7–33) and C-terminal rat Cd4 domains 3 and 4, followed by a biotinylatable peptide. The Cd4 tag was used to monitor protein folding as it is recognised by a conformation epitope-specific monoclonal antibody. For pentameric proteins a similar pTT3 derivative was used, but a pentamerisation domain of the rat cartilage oligomeric matrix protein (COMP) conjugated to β-lactamase replaced the biotinylatable peptide[47].

Constructs were transiently transfected into HEK293E cells and soluble proteins were collected from culture supernatant on day 6. Monobiotinylated proteins required cotransfection with a modified BirA biotin ligase expression plasmid, and excess free biotin was removed by dialysis. Selective labelling of target proteins in supernatants precluded purification; however, target protein activities were quantified and normalised[47]. Relative mono-biotinylated protein concentrations were determined by ELISA, using streptavidin-coated 96-well plates and mouse anti-rat Cd4 (1:1000, OX68, MCA1022, Serotec) before goat anti-mouse alkaline phosphatase (1:5000, A0168, Sigma–Aldrich) and detection with phosphatase substrate. Pentameric proteins were analysed by colorimetric β-lactamase activity assays, where nitrocefin hydrolysis by supernatant serial dilutions was monitored by measuring absorbance at 485 nm.

**Western blot.** Denatured supernatants of monomeric biotinylated proteins were resolved using NuPAGE™ Novex™ 4–12% Bis-Tris protein gels, before blotting onto nitrocellulose membrane (GE Healthcare). After blocking, membranes were incubated with streptavidin-horseradish peroxidase (1:2000, Cell Signaling Technology) for 1 h and developed with Amersham ECL Prime chemiluminescence substrate (GE Healthcare).

**Erythrocyte sources.** Human O + erythrocytes were supplied by NHS Blood and Transplant, Cambridge, UK and Research Blood Components, Boston MA, USA, and all samples were anonymised. The work complied with all relevant ethical regulations for work with human participants. The use of erythrocytes from human donors for *P. falciparum* culture and binding studies was approved by the NHS Cambridgeshire 4 Research Ethics Committee (REC reference 15/EE/0253) and the Wellcome Sanger Institute Human Materials and Data Management Committee. Chimpanzee (*Pan troglodytes*) erythrocytes were obtained from the unused fraction of blood samples taken for health testing purposes by the Animal Health Team at Chester Zoo, UK. Macaque erythrocytes were obtained from the Yerkes National Primate Research Center, Atlanta GA, USA. Cultured red blood cells were differentiated from bone marrow-derived Cd34 + haematopoietic stem cells (Lonza).

**Erythrocyte binding assays.** Monomeric biotinylated PfEBA165, PrEBA165 and PfEBA175 protein supernatants were immobilised on 0.4–0.6 μm streptavidin-coated Nile Red fluorescent microbeads (Spherotech), by incubation with gentle agitation at 4 °C for 1 h. Loaded microbeads were washed with Buffer A (HBS, 1% BSA) and incubated in an ice-cold sonicating water bath for 20 min to disrupt aggregates. Binding reactions were carried out in flat-bottomed 96-well plates, with each well containing ~4 × 10⁵ cells and loaded fluorescent microbeads, mixed at estimated cell to fluorescent bead ratio of 1:145. After 1 h at 4°C with gentle agitation, cells were washed twice in Buffer A and analysed by flow cytometry, using an LSRII cytometer (BD Biosciences) for erythrocytes and MACSQuant (Miltenyi Biotech) for cultured red blood cells (cRBCs). The gating strategy is provided in Supplementary Fig. 5. To validate binding specificity cells were treated with *Vibrio cholerae* neuraminidase (Sigma–Aldrich). Erythrocytes and cRBCs were incubated with 20 mU/ml or 66.7 mU/ml neuraminidase, respectively, with gentle mixing for 1 h at 37 °C. The enzyme was removed by three washes in incomplete RPMI (cRBC final wash used PBS pH 7.4).

**Glycan binding assays.** Recombinant pentameric β-lactamase-tagged PfEBA175, PfEBA165 and PrEBA165 proteins were screened for interactions against synthetic carbohydrate probes (GlycoTech)[22]. Biotinylated carbohydrate probes (designated

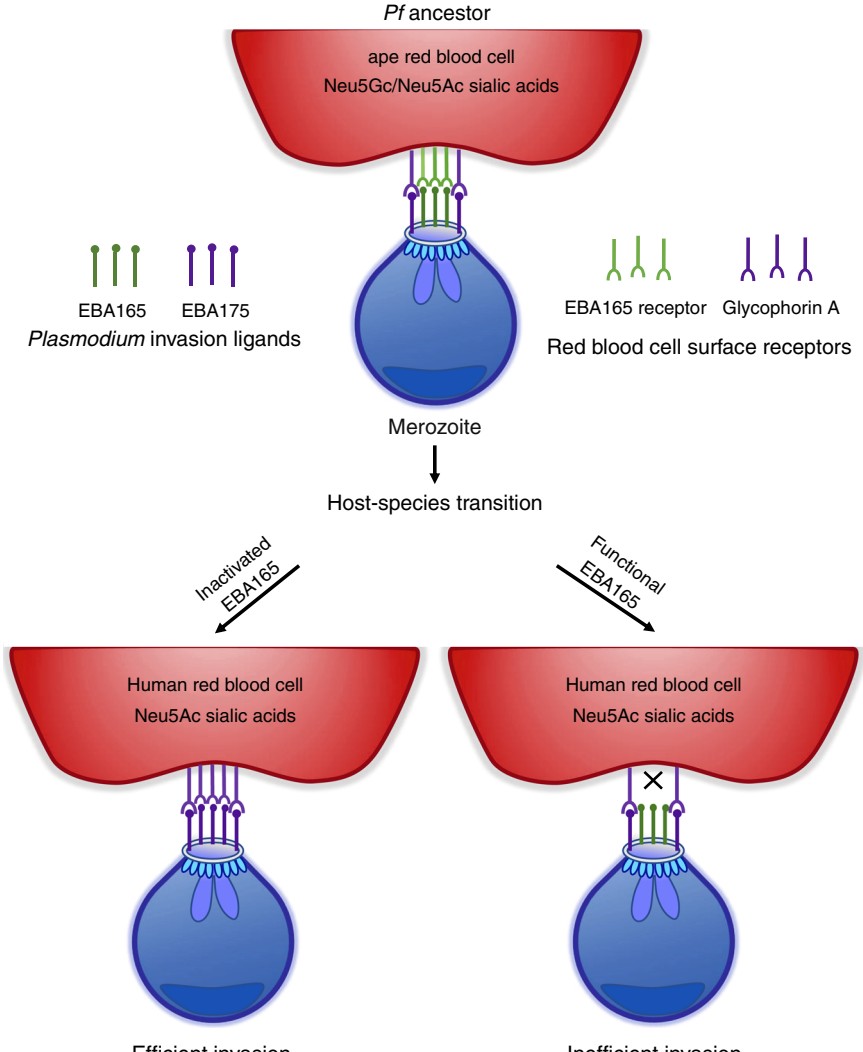

**Fig. 6** *PfEBA165* inactivation may have predisposed *P. falciparum* ancestor to invade human erythrocytes. Invasion ligands function in a partially redundant manner during erythrocyte recognition, and gene deletion studies suggest a defined hierarchy of interactions. If EBA165, which only recognises ape-specific sialic acids (Neu5Gc), was the dominant invasion ligand in the ancestor of *P. falciparum*, inactivation may have allowed PfEBA175, which is not Neu5Gc specific, to play a more dominant role in invasion (left hand arrow). This would have either predisposed parasites to more effectively invade human erythrocytes, which only expresss Neu5Ac, or increased growth after host-species transition had occurred, selecting for strains where *PfEBA165* was inactivated. By contrast ancestral parasites with a dominant and intact *PfEBA165* would still primarily rely on Neu5Gc sialic acids for invasion, and hence would invade human erythrocytes less efficiently, reducing the likelihood of species transition, or reducing growth rates once the transition had occurred

baits) were immobilised on streptavidin-coated 96-well plates by incubating at saturating concentrations for 1 h. Plates were washed three times in HBST and normalised pentameric β-lactamase-tagged proteins (designated preys) were added (100 μl/well) and incubated for 1 h. Plates were washed with HBST then HBS, before analysing the hydrolysis of nitrocefin substrate (60 μl at 125 μg ml$^{-1}$) by measuring absorbance at 485 nm.

**Cultured red blood cells (cRBCs)**. Neu5Gc was expressed on the surface of modified human erythrocytes by expressing chimpanzee *CMAH* in ex-vivo-cultured red blood cells (cRBCs)[26]. Briefly, differentiation and proliferation of bone marrow-derived CD34$^+$ hematopoietic stem cells (Lonza) were achieved in Iscove's Modified Dulbecco-based medium (Biochrom) supplemented with 5% solvent/detergent virus-inactivated human plasma (Octaplas, Octapharma), stem cell factor (R&D Systems), hydrocortisone (Invitrogen), IL-3 (R&D Systems), and erythropoietin (Amgen) according to a published protocol[48] with the following modifications. Cells were transduced on day 7 with lentivirus harbouring either the chimpanzee (*Pan troglodytes*) CMAH cDNA sequence in pLVX-Puro (Clontech) or the empty vector, pLVX-Puro. The CMAH insert was codon-optimized for human expression and synthesized with a C-terminal tobacco etch virus (TEV) cleavage site and FLAG-c-myc tags (GeneArt). Transductants were selected on 2 mg ml$^{-1}$ puromycin (Sigma–Aldrich). Drug selection was maintained until day 13 when cells were co-cultured on a murine MS-5 stromal cell layer. Cells were

replated on MS-5 stroma on day 18 and harvested on day 20 or 21. Cells were stored at 4 °C in incomplete RPMI (RPMI-1640 [Sigma–Aldrich] supplemented with 25 mM HEPES and 50 mg l$^{-1}$ hypoxanthine) until binding and flow cytometry assays.

**Flow cytometry detection of Neu5Gc and erythrocyte receptors**. Neu5Gc and erythrocyte receptor surface expression was assessed by flow cytometry. Cells were washed three times in PBS containing 0.5% Neu5Gc-free blocking agent (Siamab) and pelleted at 500 g for 4 min in 96-well plates at 5 × 10$^5$ cells/well. Cells were incubated with antibodies at the following dilutions: anti-Neu5Gc (1:5000, Siamab), phycoerythrin(PE)-conjugated anti-DARC (1:10, 130-105-683, Miltenyi Biotec), anti-CD71-PE (1:10, 130-104-151, Miltenyi Biotec), anti-BSG (1:1000, Clone MEM-M6/6, Axxora [Exbio]), fluorescein isothiocyanate-conjugated anti-glycophorin A (GPA) (1:50, Clone 2B7, 60152FI, STEMCELL Technologies) and anti-glycophorin C (GPC)-FITC (1:500, BRIC 10, sc-59183 FITC,Santa Cruz). After 1 h incubation at room temperature and three washes in blocking buffer, unstained cells and cells stained with DARC, CD71, GPA and GPC antibodies were resuspended in 100 μl PBS for flow cytometry analysis (MACSQuant; Miltenyi Biotec). Neu5Gc-stained cells were incubated in anti-chicken IgY-Alexa Fluor 488 secondary antibody (Life Technologies) at 1:1000 for 30 min at room temperature. Unstained control samples were incubated in anti-mouse IgG2a-PE (1:10, Miltenyi Biotec), anti-chicken IgY-Alexa Fluor 488 antibody or anti-mouse

IgG-Alexa Fluor 488 (1:1000, Life Technologies). Cells were washed twice before flow cytometry. Data were analysed in FlowJo 4 version 10.0.7 (Tree Star).

**P. falciparum culture and CRISPR modification**. Wild-type 3D7 *Plasmodium falciparum* strain was obtained from Malaria Research and Reference Reagent Resource Centre (www.mr4.org). *P. falciparum* strain 3D7 was cultured in RPMI-based media supplemented with 0.5% AlbuMAX II (Life Technologies), 2 mM L-glutamine and O+ human erythrocytes, using standard techniques[49]. For transfection, sorbitol-synchronised ring stage parasites (~ 10% parasitemia, 2.5% haematocrit) were electroporated with 50 μg plasmid DNA and 25 μg Donor Template (PCR product) using a Lonza 4D-Nucleofector. Briefly, 100 μl of infected erythrocytes were washed and resuspended in 100 μl of Primary Cell Nucleofector Solution P3 (Lonza) supplemented with 12.5 mM ATP and DNA for transfection. The cell mixture was split into two Nucleocuvettes and electroporated using programme CM150. Cells were placed on ice for 2 mins then transferred to pre-warmed complete media for 3 h, before replacing media and returning to standard culture conditions. At 24 h post-transfection drug selection was applied with 1.5 nM WR92210 (Jacobus Pharmaceuticals) for 6–12 days. To edit *PfEBA165* a CRISPR/Cas9 approach was used, modifying a pDC2-cam-Cas9-U6-sgRNA-hDHFR construct[50] to independently target the two frameshifts found in *PfEBA165* in *P. falciparum* strain 3D7. sgRNA-1 (sequence GATAAGTTGGCAAATGGG AGTT) and sgRNA-2 (sequence GAACATAAAGAGAAATTTCCAA) were designed to target the 5′ and 3′ *PfEBA165* frameshifts, respectively. The sgRNAs were inserted into *Bbs*I cut vector following annealing of the oligonucleotide pairs OL176-OL177 (sgRNA-1), and OL231-OL232 (sgRNA-2). To make *PfEBA165* edits, donor templates were supplied as purified PCR product. Fragments of *PfEBA165* were cloned into TOPO and modified by KAPA HiFi site directed mutagenesis to incorporate modifications for frameshift correction and silent mutations in the Cas9 cut site (Fig. 5). For transfection, primers with 2 phosphorothioate bonds at the 5′ prime end were used to PCR amplify the Donor Template. Before transfection, plasmids and PCR products were precipitated and resuspended in Primary Cell Nucleofector Solution P3 (Lonza). Transfectants were analysed by Sanger sequencing, using OL271 and OL293 to amplify ~2.1 kb fragment of *PfEBA165* outside of both donor template regions. Resulting PCR products were purified and sequenced directly. Oligonucleotide sequences are listed in Supplementary Table 2.

**Whole-genome sequencing of edited lines**. Paired-end reads were aligned to the *P. falciparum* version 3 genome sequence using bwa-mem[51] [https://arxiv.org/abs/1303.3997]. SNPs and indels were called using Freebayes[52] [https://arxiv.org/abs/1207.3907] with default parameters, other than to specify a haploid genome. Variants were filtered for an overall quality score >20 and a sequencing depth >8 in all samples. Any variants with > 5 alternate observations in the sequenced parental 3D7 strain were discarded as being the result either of poor mapping or a pre-existing mutation. The effect of the detected variants upon translated protein sequences was predicted using SnpEff[53].

**RNA preparation**. To synchronize parasites a 70% percoll gradient was used to collect late schizonts. Following 2 h incubation newly reinvaded parasites were separated with 70% percoll, and ring stages were further synchronized with 5% sorbitol. Parasites were allowed to develop until late-stage schizonts could be harvested. Total RNA was extracted using the Trizol (Gibco) method according to manufacturer's instructions. RNA quality was verified using an Agilent Bioanalyzer 2100.

**Erythrocyte invasion assays**. A two-colour flow cytometric assay was used to quantitate erythrocyte invasion by *P. falciparum*[54]. Briefly, to label target erythrocytes, cells were incubated with 10 μM CellTrace™ (Thermo Fisher Scientific) Far Red (DDAO) for 2 h at 37 °C. The suspension was washed and incubated in complete medium for 30 min, followed two further washes. Where required, enzymatic treatment of erythrocytes was performed by incubation for 1 h at 37 °C under rotation with enzymes at the following final concentrations: neuraminidase from *Vibrio cholerae* (Sigma–Aldrich) 20 mU ml$^{-1}$; trypsin (Sigma–Aldrich) or chymotrypsin (Sigma–Aldrich, Dorset, UK) 50 μg ml$^{-1}$ (low trypsin) or 1 mg ml$^{-1}$ (high trypsin and chymotrypsin). Enzymes were removed by washing with incomplete media. Invasion assays were conducted in round-bottom 96-well plates, by mixing equal volumes of 2% haematocrit suspensions of labelled erthrocytes and parasitised erythrocytes (donor culture). The final culture volume was 100 μl per well and plates were incubated for 48 h at 37 °C inside a gassed culture chamber. To quantify invasion into target erythrocytes, cultures were washed in PBS and stained with SYBR Green I (Invitrogen) DNA dye, at 1:5000 final concentration. Cells were washed with PBS before acquisition on a BD LSRII flow cytometer (BD Biosciences) with a 355 nm UV laser (20 mW), a 488 nm 20 mW blue laser, and a 633 nm red laser (17 mW). SYBR Green I was excited by a blue laser and detected by a 530/30 filter. DDAO was excited using the red laser and detected with a 660/20 filter. Data were analyzed with FlowJo (Tree Star).

**qRT-PCR**. For quantitative RT-PCR, RNA was extracted from late-stage synchronous parasites and treated with DNase I (Ambion AM1907) to remove gDNA

contamination. cDNA was prepared using the High Capacity Reverse Transcription Kit (Thermofisher 4368814) from 1 μg of total RNA using a mix of oligo dT (Ambion AM5730G) and random hexamers (Invitrogen). A LightCycler 480 (Roche) was used to assess transcript levels in reactions containing SYBR green real time reagent (Roche 04707516001), template cDNA and 0.2 μM of each primer. Relative quantification of transcripts was performed using the ΔΔCt method, with transcript levels normalized to *Pfcyp87* (stable expression throughout cell cycle) and expressed relative to *PfAMA1* (late-stage specific expression) to control for variation in RNA level (*Pfcyp87* normalisation) as well as minor differences in life-cycle stage synchronisation (normalisation to *PfAMA1*). Sequences of primer pairs targeting *Pfcyp87* (PF3D7_0510200), *PfAMA1* (PF3D7_1133400) and *PfEBA165* (Pf3D7_0424300) are shown in Supplementary Table 2.

**RNAseq**. A modified RNA-seq protocol (DAFT-seq) was used to account for the extreme AT-content of the *P. falciparum* transcriptome[55]. PolyA + RNA (mRNA) was selected using magnetic oligo-d(T) beads. Reverse transcription using Superscript III (Life) was primed using oligo d(T) primers, then second strand cDNA synthesis included dUTP. The resulting cDNA was fragmented using a Covaris AFA sonicator. A with-bead protocol was used for dA-tailing, end repair and adapter ligation (NEB) using PCR-free barcoded sequencing adaptors[56] (Bioo Scientific). After two rounds of SPRI cleanup the libraries were eluted in EB buffer and USER enzyme mix (NEB) was used to digest the second strand cDNA, generating directional libraries. The libraries were quantified by qPCR and sequenced on an Illumina HiSeq2000 to generate about 20 million reads per sample. Quality of the fastq files were assessed with the FastQC tool (Babraham Bioinformatics) and reads with Phred quality scores over 30 were used for subsequent analysis. The processed paired-end reads were mapped to the *Plasmodium falciparum* 3D7 genome (PlasmoDB, v29) using Tophat2 v2.1.1. DESeq2 was used in the R statistical environment to normalize and determine significantly differentially expressed genes with adjusted p-values ≤ 0.05 between 3D7 control and double-edited clones using default parameters (DESeq2, Wald-test[57]). Results of differential expression analysis were curated to remove genes previously shown to have significant clonally-variant expression[58] and were also characterized further using gene ontology enrichment using PlasmoDB's GO enrichment strategy tool. Data have been deposited in the European Nucleotide Archive (ENA) with accession number ERP114933 and is accessible through the SRA.

**Reporting summary**. Further information on research design is available in the Nature Research Reporting Summary linked to this article.

## Data availability

RNA-sequencing data have been deposited in the European Nucleotide Archive (ENA) with the accession number ERP114933 [https://www.ebi.ac.uk/ena/data/view/PRJEB32274], and corresponding DNA-sequencing data with accession number ERP023755 [https://www.ebi.ac.uk/ena/data/view/PRJEB21498]. The source data for Figs. 2a, 3a, 4c and Supplementary Figs 4a-d are available in the Source Data file. All other data are available from the authors upon reasonable request.

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

## Acknowledgements

We thank Marcus Lee for supplying CRISPR-Cas9 vectors and Theo Sanderson for help with whole-genome sequencing analysis. This work was supported by grants from the National Institutes of Health (R01 AI091595, R37 AI050529, R01 AI 120810, T32 AI007532, P30 AI045008) and the Wellcome Trust (206194/Z/17/Z).

## Author contributions

The study was conceived by W.R.P., B.H.H. and J.C.R. Experiments were performed by W.R.P., S.V.S., S.D., W.L., A.K., S.M. and Z.Z. Data analysis and interpretation were performed by W.R.P., S.V.S., S.D., S.U., P.M.S., G.J.W., B.H.H., M.T.D. and J.C.R. The manuscript was written by W.R.P and J.C.R., with contributions from all co-authors.

## Additional information

**Competing interests:** The authors declare no competing interests.

