## [Peer Review File · Nature Communications]

Reviewers' Comments:

Reviewer #1:

Remarks to the Author:

Proto et al. provide evidence that a specific mutation in the *Plasmodium falciparum* (malaria parasite) invasion ligand EBA165 that renders the protein into a pseudogene and is present in all isolates sequenced to date is critical for parasite invasion into human red blood cells. A functional copy of EBA165 was found by the authors in the reference genomes and field isolate DNA for all closely related ape malaria parasites (*Laverania*), suggesting the antigen is required for ape-RBC invasion. One key difference in RBC receptors between human and ape cells is the presence of both Neu5Gc and Neu5Ac sialic acid residues on ape cells and of only Neu5Ac residues on human RBCs due to disruption of the human N-glycolylneuraminic acid gene (CMAH) that converts Neu5Ac to Neu5Gc.

The authors show that recombinant ape malaria EBA165 (PrEBA165) and full-length corrected (frame shift repaired) *P. falciparum* EBA165 (PfEBA165) bound to chimpanzee RBCs and also to cultured transgenic human RBCs that expressed the ape Neu5Gc. Binding assays showed that both PrEBA165 and PfEBA165 bind to Neu5Gc sialic acid residues of ape RBCs, but not to Neu5Ac residues. These experiments convincingly show a strong link between EBA165 and binding to ape Neu5Gc labelled proteins sialoglycoproteins.

The authors then go onto repair the frameshift mutations in the *P. falciparum* 3D7 parasite line to create a functional EBA165. However, no change in growth or invasion phenotypes was observed with this transfected line. Further investigation revealed that the expression of repaired EBA165 was highly downregulated. This suggests that expression of functional EBA165 is deleterious to parasite growth, and the incorporation of the Frameshift mutation into this invasion ligand may have been a key step in *P. falciparum*'s adaption to human RBCs.

This novel finding is significant since it potentially identifies a key step in the transition of *P. falciparum*'s ape infecting ancestors into the major human pathogen it has become. The manuscript is well written and should be of interest to readers. The conclusions are based on strong evidence that has been achieved using nicely designed experiments.

Major comments

-This paper demonstrates EBA165 binding to Neu5Gc sialic acid residues (neuraminidase sensitive) using recombinant protein in human/chimpanzee RBC binding, cultured human RBCs expressing Neu5Gc residues and with a glycan array featuring both Neu5Gc and Neu5Ac residues. This is very strong binding assay data, but if the study lacks anything, it is the ability to confirm EBA165s use of Neu5Gc during merozoite invasion in growth assays, and whether expression of PfEBA165 is deleterious to parasite invasion directly. There are clear difficulties in testing this such as 3D7 parasites with a frameshift corrected EBA165 completely downregulating expression of this now functional antigen. The Duraisingh group previously showed that parasite growth could be maintained in cRBCs (Bei et al 2010). Did the authors attempt to select for genetically modified 3D7 parasites expressing functional EBA165 by growing them in cRBCs expressing CHAM and Neu5Gc on the cRBC surface? If successful, these could then be used to directly investigate EBA165 during invasion into both Neu5Gc +ve and -ve cells. If this was not attempted, could the authors comment on whether it is feasible using the current scale of cRBC cultures? Could this be done to really nail down the properties of PfEBA165 during invasion.

-In figure S4, the authors suggest that sialic acid treatment of Hs and Pt cells did not cause a change in invasion for the EBA165 frameshift corrected transfected 3D7 lines. I suggest that there may be a trend towards increased neuraminidase sensitivity (where there any statistical tests done to test for this?), particularly for S4 C, suggesting a movement towards a sialic acid dependent invasion pathway. Despite EBA165 expression dropping, this might make sense given the loss of RH4 expression. RH4 is an important non-sialic acid dependent pathway for 3D7 invasion. Loss of RH4

could switch the parasite more towards a neuraminidase sensitive, sialic acid dependent, invasion pathway. Can the authors confirm that the variation evident for the neuraminidase treatments has been tested directly and discuss the implications of loss of Rh4 expression on invasion pathways for the genetically engineered parasites.

-In figure S4, Panels A and B are in the wrong order compared to the Figure legend.

-There isn't a lot of statistical analysis of the data evident in the text, and for several graphs there is an opportunity to strengthen the comparative analysis by doing so. It is also not clear for some data sets whether the experiment was repeated to assess reproducibility or whether a single experiment with triplicate wells was done. Error bars are also not defined in all figures. This should be rectified.

-The experiments are well done and would be readily repeatable if other researchers had access to some of the rarer host cell types such as the Chimpanzee and cRBCs.

Minor comments

There is no line numbering to highlight some minor text changes. So below I past in the sentence and highlight the text that needs modifying in capitals.

- Both were unable to bind to control cRBCs (pLVX, Fig 3B), just as they had been unable to bind to mature human erythrocytes taken from circulation, but they could bind cRBCs differentiated from CMAH expressing HSCs, which differed only in the sialic acid variant that was expressed on the red blood CELL surface.

- This and other molecular studies are starting to reveal the exact steps that were required for *P. falciparum* to emerge as one of the most significant PATHOGENS in human history.

- To quantify invasion into target ERYTHROCYTES, cultures were washed in PBS and stained with SYBR Green I (Invitrogen) DNA dye, at 1:5000 final concentration.

-Recombinant PfEBA165, PrEBA165 and PfEBA175 ECTODOMAINS were expressed as secreted mono-biotinylated proteins in HEK239E cells.

Reviewer #2:

Remarks to the Author:

REVIEW of "Adaptation of *Plasmodium falciparum* to humans involved the loss of an ape-specific erythrocyte invasion ligand." by Proto et al for Nature Communications.

SUMMARY

EBA165 belongs to a family of several *Plasmodium* parasite erythrocyte binding ligands of the Laveranian group that infect humans and other apes. The EBA ligands play a role in helping the merozoite parasite stage to attach to and invade RBCs. Here Proto et al establish that all in human *P. falciparum* species tested the EBA165 gene does not produce a protein due to a 5' frame shift mutation. In Pf-like Laveranian parasites that infect African apes EBA165 does produce a functional protein that is shown to bind to certain types of Neu5Gc sugars present on ape RBCs but not on human RBCs whose proteins are decorated with Neu5Ac sugars. Recombinant PfEBA165 produced from the frame shift repaired gene is still functional and able to bind ape RBCs but not human RBCs. To show that ape sugars are important for RBC binding, stem cell derived human RBCs were transfected with the chimpanzee CMAH gene that converts Neu5Ac to Neu5Gc. PfEBA165 bound to the modified human RBCs along with an ape EBA165. When the frame shifted PfEBA165 was repaired by CRISPR to produce a functional protein the modified Pf parasites were able to grow in human RBCs but only if expression of functional PfEBA165 was greatly reduced. This indicated that expression of PfEBA165 is incompatible with growth in human RBCs and the inactivation of EBA165 must have been an important event when Pf's ancestors moved from apes into humans.

MAJOR COMMENTS

This is an excellent and fascinating paper and I can find no major criticisms of the work. It would be interesting to know the why the expression of functional PfEBA165 in Pf parasites is incompatible with growth in human RBCs. It was speculated that functional PfEBA165 might prevent other invasion pathways such as that employing PfEBA175 from working properly as shown in Fig 6. There is no evidence to support the EBA175 hypothesis but I don't think this is needed at this stage. Also, it would be interesting to know which Neu5Gc decorated ape protein(s) is the EBA165 receptor and if this is the same as that in CMAH-expressing human RBCs but that can be done in the future.

MINOR COMMENTS

1. In Fig S3 the legend states that "cRBCs expressed lower levels of DARC and CD71 than mature erythrocytes, as has been previously shown." Shown where? Unless I am missing something the graphs show CD71 is higher in cRBCs than human RBCs.
2. In the methods it states the oligo sequences are shown in Table S1 but Table S1 shows gene expression data. Table S1 is labeled as Table 1 in the table but S1 in the legend.

Response to reviewers:

REVIEWER 1: MAJOR COMMENTS

-This paper demonstrates EBA165 binding to Neu5Gc sialic acid residues (neuraminidase sensitive) using recombinant protein in human/chimpanzee RBC binding, cultured human RBCs expressing Neu5Gc residues and with a glycan array featuring both Neu5Gc and Neu5Ac residues. This is very strong binding assay data, but if the study lacks anything, it is the ability to confirm EBA165s use of Neu5Gc during merozoite invasion in growth assays, and whether expression of PfEBA165 is deleterious to parasite invasion directly. There are clear difficulties in testing this such as 3D7 parasites with a frameshift corrected EBA165 completely downregulating expression of this now functional antigen. The Duraisingh group previously showed that parasite growth could be maintained in cRBCs (Bei et al 2010). Did the authors attempt to select for genetically modified 3D7 parasites expressing functional EBA165 by growing them in cRBCs expressing CHAM and Neu5Gc on the cRBC surface? If successful, these could then be used to directly investigate EBA165 during invasion into both Neu5Gc +ve and -ve cells. If this was not attempted, could the authors comment on whether it is feasible using the current scale of cRBC cultures? Could this be done to really nail down the properties of PfEBA165 during invasion.

We thank the reviewer for their question, which we agree is important and had previously considered a great deal before submission of the manuscript. Unfortunately we believe that the experiment is simply technically impossible, at least at this time. There are three hurdles - the inefficiencies of transfection in *P. falciparum*, the length of time that such selection experiments usually take, and the low volumes of cRBCs that are generated in vitro. Transfection efficiencies in *P. falciparum* are such that we believe that only a handful of parasites are successfully transfected in each experiment, based on the 5-8 weeks it can take for parasites to reappear in culture after transfection. Selection for different invasion pathways have only ever been carried out successfully in one strain to our knowledge (and the submitting authors on this manuscript have tried several other strains without success), and in this case it also takes several weeks for parasites to appear. These two factors together require a long period of continuous culture, almost certainly on the scale of months. Generating each batch of cRBCs takes considerable resources both in time (c. 3 weeks for each batch) and materials, and produces 10ul of CMAH cRBCs (1×10^8 cells) at best before filtration to remove the very young nucleated cells. At ~20% recovery post-filtration, this leaves 2×10^7 cells. This number of cells is sufficient for short-term invasion and growth assays, such as those carried out in this manuscript and others previously published but is inadequate to support even small-scale continuous parasite culture. Producing a consistent supply of CMAH cRBCs to supplement a culture over a period of months is therefore unfortunately just not possible at this time. We completely agree that the idea is an important one however, and we have now added text to the Discussion explaining the concept and technical limitations.

-In figure S4, the authors suggest that sialic acid treatment of Hs and Pt cells did not cause a change in invasion for the EBA165 frameshift corrected transfected 3D7 lines. I suggest that there may be a trend towards increased neuraminidase sensitivity (where there any statistical tests done to test for this?), particularly for S4 C, suggesting a movement towards a sialic acid dependent invasion pathway. Despite EBA165 expression dropping, this might make sense given the loss of RH4 expression. RH4 is an important non-sialic acid dependent pathway for 3D7 invasion. Loss of RH4 could switch the parasite more towards a neuraminidase sensitive, sialic acid dependent, invasion pathway. Can the authors confirm that the variation evident for the neuraminidase treatments has been tested directly and discuss the implications of loss of Rh4 expression on invasion pathways for the genetically engineered parasites.

We thank the reviewer for contributing this new insight. As predicted by the reviewer, the increased sensitivity of the double edited strains to neuraminidase treatment during invasion is statistically significant, and is consistent with the RNAseq data which shows down-regulation of PfRH4. We have modified the figures and text to reflect this and have provided additional consideration on implications on the loss of RH4 in the Discussion.

-In figure S4, Panels A and B are in the wrong order compared to the Figure legend.

This error has now been corrected.

-There isn't a lot of statistical analysis of the data evident in the text, and for several graphs there is an opportunity to strengthen the comparative analysis by doing so. It is also not clear for some data sets whether the experiment was repeated to assess reproducibility or whether a single experiment with triplicate wells was done. Error bars are also not defined in all figures. This should be rectified.

We have made several changes to help clarify how replicates were conducted and make clear where statistical tests have been applied. Figure legends have been modified to clarify the experimental repeats performed and to define all error bars. In addition, the outcomes of statistical test have also been added to the appropriate data (see Figs. 4C, Supplementary Figs 4B and 4C).

-The experiments are well done and would be readily repeatable if other researchers had access to some of the rarer host cell types such as the Chimpanzee and cRBCs.

We're pleased that experiments have been designed and presented in a way that will help other researchers in the field. Obtaining chimpanzee RBCs for even these small number of experiments was very challenging, for good reason, and we are grateful to the support of our colleagues at Chester Zoo who made it possible.

MINOR COMMENTS

There is no line numbering to highlight some minor text changes. So below I past in the sentence and highlight the text that needs modifying in capitals.

- Both were unable to bind to control cRBCs (pLVX, Fig 3B), just as they had been unable to bind to mature human erythrocytes taken from circulation, but they could bind cRBCs differentiated from CMAH expressing HSCs, which differed only in the sialic acid variant that was expressed on the red blood CELL surface.

- This and other molecular studies are starting to reveal the exact steps that were required for P. falciparum to emerge as one of the most significant PATHOGENS in human history.

- To quantify invasion into target ERYTHROCYTES, cultures were washed in PBS and stained with SYBR Green I (Invitrogen) DNA dye, at 1:5000 final concentration.

-Recombinant PfEBA165, PrEBA165 and PfEBA175 ECTODOMAINS were expressed as secreted mono-biotinylated proteins in HEK239E cells.

We thank the reviewer for their careful reading of the manuscript and have corrected all typographical errors.

**REVIEWER 2:
MAJOR COMMENTS**

This is an excellent and fascinating paper and I can find no major criticisms of the work. It would be interesting to know the why the expression of functional PfEBA165 in Pf parasites is incompatible with growth in human RBCs. It was speculated that functional PfEBA165 might prevent other invasion pathways such as that employing PfEBA175 from working properly as shown in Fig 6. There is no evidence to support the EBA175 hypothesis but I don't think this is needed at this stage. Also, it would be interesting to know which Neu5Gc decorated ape protein(s) is the EBA165 receptor and if this is the same as that in CMAH-expressing human RBCs but that can be done in the future.

We are pleased the reviewer found the paper and model presented of interest. The experiment suggested would certainly have value, however, identifying host targets of parasite invasion ligands remains a considerable challenge in the field, even with the addition of new screening tools such as AVExis. In addition in this case, the limited supply of chimp erythrocytes and cRBC make biochemical approaches next to impossible. Despite these challenges, we absolutely agree that identifying the chimpanzee receptor for EBA165 is a topic of some interest, and it is one that we will pursue in the future.

MINOR COMMENTS

1. In Fig S3 the legend states that "cRBCs expressed lower levels of DARC and CD71 than mature erythrocytes, as has been previously shown." Shown where? Unless I am missing something the graphs show CD71 is higher in cRBCs than human RBCs.

We thank the reviewer for their observation, which is absolutely right. We have corrected the figure legend to make it clear that there is significantly higher expression of CD71, but lower expression of DARC on cRBCs compared to mature erythrocytes. This is consistent with what was have reported previously (see Dankwa et al. 2016, Nature Communications, and Giarratana et al. 2011. Blood, for CD71 expression) for cRBCs, and probably relates to the slightly younger age of such cells. This point has also now been made in the manuscript

2. In the methods it states the oligo sequences are shown in Table S1 but Table S1 shows gene expression data. Table S1 is labeled as Table 1 in the table but S1 in the legend.

The oligo table has now been added to the Supplementary Information file as Supplementary Table 2, and the text has been modified to correct the error. The incorrect Table 1 label from Supplementary Table 1 has also been removed.

Reviewers' Comments:

Reviewer #1:

Remarks to the Author:

A second review of the manuscript reveals it is in good shape, except for Supplementary Fig 4 B and C. Inclusion of the number of experimental replicates reveals that each data set was only done once. This is understandable for Pt (Chimpanzee) blood experiments. But the data for Human RBCs is therefore weakened. Quite simply, probably shouldn't do statistics on a single experiment. If additional experiments are not to be done, I suggest talking about this in terms of trends. I also suggest modifying the figures using the following.

-Create a separate figure for Hs +/- neuraminidase and use the data available in Figure 4C to create a figure establishing the trend across two biological replicates. This would be stronger than having two separate trends and is clearer than comparing between Supplementary 4B and 4C to see repeat Hs experiments with Neuraminidase.

-Follow that with the Pt +/- neuraminidase figure only.

-Finish with the Hs +/- remaining enzyme treatments (I don't think having data for YB4 adds anything here).

Any additional data available that could increase the number of biological replicates would be of benefit.

Given the limited importance of these observations in the overall manuscript, I feel that modification of the figures, inclusion of any available repeat data and discussing the data in terms of trends (rather than as statistically significant) would be sufficient. If the authors want to strengthen this data with repeat experiments for Hs RBCs, that's up to them.

REVIEWERS' COMMENTS:

Reviewer #1 (Remarks to the Author):

A second review of the manuscript reveals it is in good shape, except for Supplementary Fig 4 B and C. Inclusion of the number of experimental replicates reveals that each data set was only done once. This is understandable for Pt (Chimpanzee) blood experiments. But the data for Human RBCs is therefore weakened. Quite simply, probably shouldn't do statistics on a single experiment. If additional experiments are not to be done, I suggest talking about this in terms of trends. I also suggest modifying the figures using the following.

-Create a separate figure for Hs +/- neuraminidase and use the data available in Figure 4C to create a figure establishing the trend across two biological replicates. This would be stronger than having two separate trends and is clearer than comparing between Supplementary 4B and 4C to see repeat Hs experiments with Neuraminidase.

-Follow that with the Pt +/- neuraminidase figure only.

-Finish with the Hs +/- remaining enzyme treatments (I don't think having data for YB4 adds anything here).

Any additional data available that could increase the number of biological replicates would be of benefit.

Given the limited importance of these observations in the overall manuscript, I feel that modification of the figures, inclusion of any available repeat data and discussing the data in terms of trends (rather than as statistically significant) would be sufficient. If the authors want to strengthen this data with repeat experiments for Hs RBCs, that's up to them.

We thank the reviewer for their detailed and helpful reading of the revised manuscript, and we have adjusted Supplementary Figure 4 exactly as requested.